

# High-resolution shear wave reflection seismics as tool to image near-surface subrosion structures — a case study in Bad Frankenhausen, Germany

Sonja Wadas[1], Ulrich Polom[1], and Charlotte Krawczyk[1,2]

[1]Leibniz Institute for Applied Geophysics, Stilleweg 2, D-30655 Hannover, Germany
[2]now GFZ German Research Centre for Geosciences, Telegrafenberg, D-14473 Potsdam, Germany

*Correspondence to:* Sonja Wadas (sonja.wadas@liag-hannover.de)

**Abstract.** Subrosion is the subsurface leaching of soluble rocks that results in the formation of depression and collapse structures. This global phenomenon is a geohazard in urban areas. To study near-surface subrosion structures four shear-wave reflection seismic profiles with a total length of ca. 332 m were carried out around the famous leaning church tower of Bad Frankenhausen in northern Thuringia, Germany, which shows an inclination of 4.93° from the vertical. Most of the geological underground of Thuringia is characterized by soluble Permian deposits, and the Kyffhäuser-Southern-Margin Fault is assumed to be a main pathway for water to leach the evaporite. The seismic profiles were acquired with the horizontal micro-vibrator ELVIS developed at LIAG and a 72 m long landstreamer equipped with 72 horizontal geophones. The high-resolution seismic sections show subrosion-induced structures to a depth of ca. 100 m and reveal five features associated with the leaching of Permian deposits: (1) lateral and vertical varying reflection patterns caused by strongly heterogeneous strata, (2) discontinuous reflectors, small offsets and faults, which show the underground is strongly fractured, (3) formation of depression structures in the near-surface, (4) diffractions in the unmigrated seismic sections that indicate an increased scattering of the seismic waves, (5) varying seismic velocities and low-velocity zones that were presumably caused by fractures and upward-migrating cavities. A previously undiscovered southward-dipping, listric normal fault was also found, located northward of the church. It probably serves as a pathway for water to leach the Zechstein formations below the church and causes the tilting of the tower. This case study shows the potential of horizontal shear-wave reflection seismics in imaging near-surface subrosion structures in an urban environment with a horizontal resolution of less than 1 m in the uppermost 10–15 m.

## 1 Introduction

Subrosion, the underground leaching of soluble rocks, is a global phenomenon and the entire process is not yet well understood. It requires the presence of soluble rocks (e.g. evaporites), unsaturated water (e.g. groundwater and meteoric water), and fractures or faults that enable the unsaturated water to circulate in order to create subsurface cavities (Smyth, 1913; Martinez et al., 1998; Galloway et al., 1999; Lauritzen & Lundberg, 2000; Yechieli et al., 2002). Depending on the leached material and the parameters of the subrosion process, especially the dissolution rate (Cooper, 1986), different kinds of structures may evolve in the subsurface. The two main types are (1) collapse and (2) depression structures (for a detailed classification sees



Waltham et al. (2005); Gutiérrez et al. (2008). Strong dissolution creates sinkholes with diameters up to several tens of meters due to the development of large underground cavities and the subsequent collapse of overlying deposits (Davies, 1951; White & White, 1969). A low dissolution rate results in sag depression structures (Beck, 1988).

Subrosion by itself is a natural process, but it can be influenced by anthropogenic factors, such as mining (Brady & Brown, 2006), manipulation of the aquifer system and/or the groundwater flow (Bell, 1988), and the extraction of saline water (Getchell & Muller, 1995). Especially the formation of sinkholes is a dangerous geohazard if they occur in urban areas, where they can then lead to building and infrastructure damage and life-threatening situations (O'Connor & Murphy, 1997; Waltham, 2002; Waltham & Lu, 2007; Parise, 2011a; Parise & Lollino, 2011b). Because of the constant increase in the world's population and rapid growth of urban areas towards zones affected by subrosion, detailed knowledge of these structures and their generation process is increasingly in demand.

A few studies have dealt with the understanding of the processes and the imaging of subrosion structures e.g. cavities, collapse sinkholes and depressions (Waltham et al., 2005). To monitor sinkhole development, ortho-rectified aerial photos and differential field GPS are used (Yechieli et al., 2002; Abelson et al., 2006), while the vertical displacement is detectable by radar interferometry (Carpenter et al., 1998; Baer et al., 2002; Abelson et al., 2003) and plane-table leveling (Scholte, 2011). Gravimetric methods are suitable for the detection of cavities and mass movement (Arzi, 1975; Neumann, 1977; Butler, 1984; Benson et al., 1995; Jahn et al., 2014), but they also deliver information about possible cavity fills, as geoelectric (Dutta et al., 1970; Militzer et al., 1979; Bataynek & Al-Zoubi, 2000; Miensopust et al., 2015) and geomagnetic (Bosch & Müller, 2001) methods can also do. Analogue and digital mechanical modeling of the development and the propagation of collapse sinkholes is also a valuable tool (Sloan, 1998; Abdulla & Goodings, 1996; Tharp, 1999; Augarde et al., 2003). Seismic (Steeples et al., 1986; Evans et al., 1994; Bolger et al., 1995; Shtivelman et al., 2002; Krawczyk et al., 2012) and ground-penetrating radar (GPR) (Kaspar & Pecen, 1975; Ulriksen, 1982; Bolger et al., 1995; Forkmann, 1997; Bataynek et al., 2002; Miensopust et al., 2015) can deliver an image of the underground and provide physical attributes. Most of the previous case studies were carried out in rural areas, since especially geophysical methods based on electric and electromagnetic principles are often affected in urban areas by strong electromagnetic noise and the presence of ferrous materials (Bosch & Müller, 2001). Investigations using boreholes (Yechieli et al., 2002; Miller et al., 2009) are also not applicable at the most urbanized locations due to the densely built-up areas, strict approval procedures and high costs (Schmidt, 2005). Reflection seismics is widely used for mining exploration purposes (Ziolkowski & Lerwill, 1979), but also in the context of geohazards such as subrosion (Miller & Millahn, 2006; Miller & Steeples, 2008). Several studies of reflection seismics in karst regions exist, but again mostly in rural areas (e.g. Evans et al., 1994; Odum et al., 1999; Miller & Millahn, 2006; Miller & Steeples, 2008; Keydar et al., 2012), whereas investigations in urban areas are sparse, especially using shear-wave reflection seismics (Krawczyk et al., 2012). This method delivers high-resolution images of the near-surface, even in urbanized regions (Polom et al., 2010; Krawczyk et al., 2013). Compared to for exmaple GPR, it is not affected by saline water, which is the case in regions with e.g., Permian deposits (Annan, 2008). This is also the case in the study area in Bad Frankenhausen, located in Thuringia in Germany.

Thuringia has a widespread sinkhole problem because of Permian deposits close to the surface that are exposed to natural and man-made subrosion processes. One of the most famous examples for the destructive consequences of subrosion is the leaning




church tower in the medieval city center of Bad Frankenhausen. To date, geophysical methods that image the near-surface down to ca. 100 m depth with high-resolution, are still missing. Here, the application of shear wave reflection seismics using equipment and surveying configurations adapted for urban areas may fill this gap. This is tested for the specific location in Bad Frankenhausen, with four shear wave reflection seismic profiles that were carried out around the leaning church tower. This

paper shows the capabilities of high-resolution shear-wave reflection seismics to detect and characterize subrosion-controlled unstable zones and structures.

## 2   Geological setting

The city of Bad Frankenhausen is located in northern Thuringia in Germany at the southern border of the Kyffhäuser hills (Fig. 1), a small range of hills in Germany, bounded by the Harz mountain range to the north and the Thuringian Basin to the

south (for an entire regional overview see Seidel (2003)). The N–S extension of the Kyffhäuser hills is ca. 6 km and the W–E extension ca. 13 km, with the highest peak being at 473 m a.s.l. The surrounding areas have a mean altitude of 150 m a.s.l.

The Kyffhäuser hills are bounded by several faults. To the north, south of the city of Kelbra, is the NW–SE trending Kyffhäuser-Northern-Margin Fault, which belongs both to the Kyffhäuser-Crimmitschau Fault Zone and the Kelbra Fault Zone. The fault has an offset of ca. 600 m in the central part and ca. 250 m near the tips of the fault (Schriel & Bülow, 1926a, b;

Franzke et al., 1986). The northward-dipping W–E trending Kyffhäuser-Southern-Margin Fault (KSM Fault) is located south of the Kyffhäuser hills in the northern part of Bad Frankenhausen (Schriel & Bülow, 1926a, b). Other large fault zones of this region are the Kelbra Fault Zone, the Finne-Gera-Jachymov Fault Zone and the Hornburger Fault (Katzung & Ehmke, 1993; Puff, 1994; Seidel, 1998).

The sediments in the south of the low hill range and north of Bad Frankenhausen are deposits from the Zechstein Sea, which

was an epicontinental ocean during the Permian. Due to sea-level changes, conglomerates, carbonates, sulfates, and salt were cyclically deposited (Richter & Bernburg, 1955). The main marine formations, in order from stratigraphically lower to higher, are Werra, Staßfurt and Leine, with the first two being the most common in the research area. The Werra Formation consists of anhydrite, limestone, copper shales, and conglomerates (Schriel & Bülow, 1926a, b). Anhydrite and gypsum layers are also part of the Staßfurt Formation (Schriel & Bülow, 1926a, b; Kugler, 1958; Reuter, 1962).

The marine sedimentation phase was followed by the terrestrial sedimentation phase of the Triassic Buntsandstein with claystones, sandstones and shales, which are only found outside the low hill range at isolated spots, whereas Muschelkalk and Keuper are found a few kilometers south of Bad Frankenhausen. Cretaceous and Jurassic deposits are not found in the entire area (Schriel & Bülow, 1926a, b; Beutler, 1995; Knoth et al., 1998; Beutler & Szulc, 1999).

During the Early Tertiary, the northern part of the Kyffhäuser was uplifted again (Freyberg, 1923) and tilted, resulting in

a 300 m fault throw on the northern margin and a southward-dipping terrain. Therefore, the low hill range is described as a half-horst as is the Harz Mountain (Katzung & Ehmke, 1993; Puff, 1994; Seidel, 1998). Tertiary deposits are exposed only at isolated locations at the southern and western margins (Schriel & Bülow, 1926a, b). In contrast the Quaternary sediments with e.g., glacial till, detritus and loess cover a large area (Schriel & Bülow, 1926a, b; Kahlke, 1975–1990).



The entire region south of the Kyffhäuser is affected by subrosion (Schriel & Bülow, 1926a, b; Reuter, 1962)(Fig. 2). The accumulation of subrosion structures south of the Kyffhäuser is the result of the combination of soluble rocks (Zechstein Formations) in the subsurface and their contact with fresh water from the southward-draining hill range that ascends alongside the KSM Fault (Reuter, 1962). The presence of salt springs due to salt water ascension and the occurrence of numerous

sinkholes and depressions at the surface are indicators of soluble rocks near the surface like the Zechstein Formations (Kugler, 1958; Reuter, 1962). Especially these formations contain many fractures and faults, which disturb the mechanical integrity of the subsurface. Additionally the fractures and faults serve as pathways for meteoric and groundwater. The dissolution probably occurs along these features, as shown by a study of Kaufmann (2014) who investigated three research areas in the vicinity of the Harz Mountains in Germany with geophysical surveys using gravimetric, electric and magnetic methods and numerical

modelling.

The Barbarossa Cave located ca. 5 km westward of Bad Frankenhausen is situated in anhydrite of the Werra Formation and is proof of the leaching processes that occurred in the geological past (Steinmüller & Siegel, 1963). One of the oldest sinkholes of the region is called Quellgrund, first mentioned in records in 998, and located in the medieval center of Bad Frankenhausen (Fig. 3a). The largest sinkhole of the Kyffhäuser region is the so called 'Äbtissinnen Grube' between Rottleben and Bad

Frankenhausen. It probably developed in the 16[th] Century and has a diameter of $160 \times 120$ m and a depth of ca. 40 m. Many other subrosion features document the continuing subrosion until today. The most recent sinkholes are found in the north-east of Rottleben directly next to the Äbtissinnen Grube and Bilzingsleben a few km south of Bad Frankenhausen (Fig. 3b). The most famous subrosion phenomenon is the leaning church tower (Fig. 3c) in the medieval center of Bad Frankenhausen, north-east of the Quellgrund sinkhole (Reuter, 1962). It is estimated that the tower started tilting in 1640 and its current inclination is

4.93° from the vertical (Scheffler & Martienßen, 2013), which exceeds that of the leaning tower of Pisa at 3.97°. Additionally, almost all buildings in the area around the leaning tower have cracks. Repeated leveling surveys indicate a constant subsidence of the whole area (Scholte, 2011).

Surveying around the leaning church tower is challenging regarding geophysical investigations. Since the church is located in the medieval center of the town the area is densely built-up. In addition the topography is strongly alternating and different

soil conditions are present at the surface.

## 3 Seismic survey

Four shear wave reflection seismic profiles were carried out around the leaning church tower of the Oberkirche in Bad Frankenhausen (Fig. 4, 5 & 6). Profile S1 was carried out on unpaved ground beside the church and crossed over remains of walls of the medieval city. Along the profile the topographic elevation decreases from 155.2 m to 146.6 m from NE to SW

(Fig. 5a,b). Profile S2 runs to the east of the leaning tower on unpaved ground and on a cobbled street known as Schwedengasse. As an additional challenge stairs were crossed with this profile (Fig. 5c,d). The topographic elevation alongside profile S2 increases from 148.3 m to 157.4 m from south to north. Profile S3 was carried out on unpaved ground and is located on a ca. 0.5 m narrow path behind the church. The topographic elevation decreases from 151.2 m to 148.3 m from SE to NW



(Fig. 5e,f). Profile S4 investigated the subsurface below the street Am Schlachtberg, which is surfaced by asphalt and concrete. The topographic elevation decreases from 158.3 m to 147.8 m from north to south (Fig. 5g,h). To meet the requirements of this challenging investigation area the equipment and the configuration used for the shear wave reflection seismics had to be adapted by splitting the landstreamer.

5     For the seismic surveying a horizontal micro-vibrator as source (Fig. 6) and horizontal geophones as receivers were used in SH configuration. This source-receiver combination reduces converted waves and enables straight forward data processing. The slower seismic velocities of SH waves enable images of higher resolution than using P waves (Dasios et al., 1999; Inazaki, 2004). A further advantage is the autonomous suppression of surface Love waves, which occurs if the first subsurface layer is of higher seismic velocity than the second layer, which is often the case on paved or compacted roads.

10     The electro-dynamic micro-vibrators ELVIS 6 and ELVIS 7 (Fig. 6) are basically comparable and are used to generate horizontally-polarized shear waves (Polom, 2003; Druivenga et al., 2011; Krawczyk et al., 2012). ELVIS 7 was used due to technical problems with ELVIS 6. The most important advantage of these small sources is the usability in urban areas (Hoffmann et al., 2008; Krawczyk et al., 2013), such as the medieval center of Bad Frankenhausen, which is extremely built-up. Secondly, an enhanced signal-to-noise ratio (S/N ratio) due to the correlation of the sweep and the recorded data, which 15 compresses the time-stretched signal to a short wavelet comparable to an impulse signal and the noise outside of the sweep frequency range is also reduced (Yilmaz, 2001).

The micro-vibrator enables good ground coupling even in areas difficult to access due to the small base plate and a total weight of ca. 95 kg, which can be increased by a person sitting on top of the micro-vibrator (Krawczyk et al., 2012). Coupling to the ground is important, because the maximum investigation depth depends partly on the emitted energy; other important 20 factors are e.g., the thickness of the weathering layer, the scattering of the seismic wave and the number and size of fractures (Brückl et al., 2005).

To record the reflected seismic shear waves, 72 horizontal geophones of type SM-6 (Input/Output Inc., 1999) attached to a landstreamer were used (Krawczyk et al., 2012). The landstreamer was mainly adapted for near-surface reflection seismic profiling on paved or compacted ground and uses a fixed geophone spacing of 1 m. Typically, the streamer is towed by a car 25 that contains the recording system (Krawczyk et al., 2012). The trigger signal was emitted simultaneously with the start of the sweep signal to begin the digital recording. Both, trigger and sweep were sent via cable from a generator developed at LIAG (Krawczyk et al., 2012, 2013).

The four shear-wave reflection seismic profiles (total length of ca. 332 m) were gathered in July 2014. Since the target depth was ca. 80 m and due to the limited space at the surface, a fixed receiver-configuration with only the source moving forward 30 was used. Prior to the survey the landstreamer was adapted for the limited space in Bad Frankenhausen and separated into three parts for manual handling, each containing 24 geophones. The vibration point distance was 2 m, 4 vibrations were excited at each vibrator location using alternating polarities. Therefore the mean CMP-fold of the profiles results in an average of 18 seismic traces per CMP. The sweep frequency was 20 to 160 Hz and the sweep had a length of 10 s (Table 1) and the record length was 12 s. Each profile was geodetically surveyed using an electronic tachymeter referenced by DGPS-based fix points.



## 4 Data processing

The data processing contained two main steps, first, the preprocessing and second, the post-geometry processing ((Table 2); for detailed explanations regarding the processing algorithms see Hatton et al. (1986); Lavergne (1989); Baker (1999); Klingen (2001); Yilmaz (2001); Brückl et al. (2005); Stark (2008); Zhang et al. (2008); Fatima et al. (2010).

The first step of data preprocessing was to check the geodetic coordinates for induced DGPS errors caused by limited satellite access. If the derivation was $\geq 1\,\mathrm{cm}$ the coordinates were manually corrected. The mean errors were $5\,\mathrm{cm}$ for S1, $8\,\mathrm{cm}$ for S2, $2\,\mathrm{cm}$ for S3, and $5\,\mathrm{cm}$ for S4. After loading the data into the processing software a visual examination of each single record was done for quality assessment and to remove noisy traces (Fig. 7). The following vibroseis correlation correlated the pilot sweep with the recorded traces in order to compress the time-stretched signal to a short wavelet such as an impulsive signal.

Afterwards the survey geometry was installed and a crooked-line binning using a $0.5\,\mathrm{m}$ bin interval was applied.

The first steps of the post-geometry processing sequence were amplitude scaling and frequency filtering to enhance the reflection response and to attenuate noise to improve the resolution and the data quality. To accomplish that an automatic gain control (AGC, $220\,\mathrm{ms}$), a bandpass filter ($15/17\,\mathrm{Hz}$–$163/165\,\mathrm{Hz}$) and amplitude normalization were applied to our data. Subsequently, the four records of every vibrator location were vertically stacked to improve the Signal-to-Noise ratio (S/N

ratio), by reducing the statistically-distributed noise and amplification of the seismic response. In this processing stage the records of the four seismic datasets needed different handling because of varying data quality, caused by the different surface conditions. In the proximity of the source high-frequency noise and harmonic distortions caused by impaired source coupling were observed in the records in all profiles. S1 and S2, which have a noticeable change in the topographic elevation of almost $10\,\mathrm{m}$, were carried out mostly on unpaved soil and on cobblestone. As a consequence the coupling of the base plate to the

surface was often hampered, resulting in partly strong harmonic distortions (Fig. 7). S3 was also carried out on unpaved soil, but the source could be balanced better leading to better ground coupling of the base plate. S4 was surveyed on a sloping terrain, but in contrast to S1 and S2 the surface was paved by asphalt and concrete, leading to less harmonic distortions due to better ground coupling of the source. Besides the vertical stack the S/N ratio was additionally improved by individually muting the data parts of each profile that were irrelevant for the following processing steps using a top mute.

The following processing steps were performed iteratively. Since most of the reflection signals were covered by noise and harmonic distortions the top mute was followed by a frequency-wavenumber filter (FK Filter), which eliminated the disturbing effects of the noisy frequencies and the harmonic distortions (Fig. 8). For each of the four datasets an individual FK filter was applied followed by a bandpass filter of $15/17\,\mathrm{Hz}$–$85/87\,\mathrm{Hz}$, because iterative frequency analyses revealed that the reflection signals contain mostly frequencies below $85\,\mathrm{Hz}$. To prepare the data for the next processing steps, the datasets were sorted from

shot domain sort to CMP domain sort. The analysis intervals were $5\,\mathrm{m}$ or $10\,\mathrm{m}$ in order to capture the lateral velocity variations. Using the Normal Move-Out correction the reflection hyperbolas were corrected to get zero-offset travel times. Residual statics correction reduced the inaccuracies at the near-surface. Afterwards the CMPs were stacked and the first seismic sections were created. After generating the seismic time sections of the four SH-wave profiles, they were examined in detail in order to iteratively improve the velocity analysis and the data processing (Fig. 9a). Due to some remaining noise a frequency analysis



was conducted on the stacked data and as a result a Bandpass Filter was applied (Fig. 9b). The final frequency bandwidths were 17/19 Hz–68/70 Hz for S2 and 17/19 Hz–73/75 Hz for S1, S3 and S4, although the main parts of the reflection signals were between ca. 20 and 60 Hz. In combination with the application of individual Spectral Balancing applied on each of the four profiles the resolution was improved (Fig. 9c). Afterwards a FD-migration processing was applied to shift the reflectors towards

their correct position and to remove diffraction signatures. Finally the seismic sections were converted into depth sections to enable a lithological correlation with five wells located near the leaning church tower.

## 5 Structural imaging and Interpretation

In the following, reflection patterns, seismic velocities and interpretations of the seismic sections are described (Figs. 10, 11). The seismic attributes, amplitude and continuity, were used for the analysis of the reflection pattern in the unmigrated

time sections (Fig. 10a). Profile S1 shows reflectors of high amplitudes from the surface down to 100 ms TWT, especially between 20 m and 60 m horizontal distance (Fig. 10a, S1). The underlying patterns down to 1200 ms TWT are characterized by discontinuous reflectors with small offsets and vertical differences of up to ca. 60 ms TWT (ca. 1–10 m) (e.g. between 60 m and 80 m distance and 200–600 ms TWT), and by a partly weaker reflection pattern that is recognizable by lower amplitudes. Dipping reflectors are also present, particularly in the near-surface (e.g. between 50 m and 80 m distance and 50–300 ms

TWT). Besides these features, diffractions are present, especially in the area below 600 ms TWT. In section S2 the highest amplitudes are found in the uppermost 100–150 ms (Fig. 10a, S2). Down to 1200 ms TWT the seismic section S2 shows the same characteristics as S1 with offsets that result in discontinuous reflectors (e.g., between 40 m and 80 m distance and 200– 600 ms TWT). Shallow-dipping reflectors at the near-surface (e.g. between 30 m and 80 m distance and 10–200 ms TWT) and diffractions, especially below 400 ms TWT are also present. Section S3 is comparable with S1 and S2 because the reflectors are

strongly discontinuous (e.g. between 40 m and 70 m distance and 150–600 ms TWT). The highest amplitudes are found in the uppermost 200 ms TWT (Fig. 10a, S3). Diffractions, which are not as pronounced as in sections S1 and S2, are visible below 500 ms TWT. Some dipping reflectors are present in the first 200 ms TWT between 40 m and 60 m distance. The reflection pattern of S4 is not comparable with the other profiles, because in contrast, the reflectors are more continuous, of higher amplitude, and most of them dip southward (e.g., between 15 m and 60 m distance and 200–600 ms TWT). No diffractions

occur in the lower half of the profile (Fig. 10a, S4).

Six features are similar for all profiles. Firstly, the reflection patterns show lateral variations. Secondly, the highest amplitudes are found in the uppermost 150–200 ms TWT, at least in the sections S1, S2 and S3, and thirdly, there is a partly weak reflection pattern below 200 ms TWT. Fourthly, discontinuous reflectors with small offsets that have vertical differences of ca. 1 m to 10 m. Fifthly, the downward-dipping reflectors located at the subsurface form depression-like structures. Sixthly, diffractions

are visible in the sections S1, S2 and S3.

One of the key factors in obtaining a good image of the underground is the generation of a proper velocity field (Fig. 10b). The shear-wave interval velocities $V_{INT}$ of the seismic sections S1, S2 and S3 do not constantly increasing with depth, rather low-velocity zones are present in the near-subsurface. This results in lateral and vertical velocity variations, especially in the



uppermost 1000 ms TWT. In general, the shear wave velocities range between ca. 160 m s$^{-1}$ and 580 m s$^{-1}$. The velocity field of S1 shows an increase of the velocity from 0 to 100 ms TWT with $V_{INT}$ values of 150 m s$^{-1}$ to 270 m s$^{-1}$ (Fig. 10b, S1). From 50 m to 80 m distance and 100 to 300 ms TWT a low-velocity zone is visible with values of 100 to 270 m s$^{-1}$. West of this, and in the same time interval, velocity values of 270 m s$^{-1}$ to 300 m s$^{-1}$ were determined. Below this zone the velocities

almost constantly increase with depth to the highest observed velocity of ca. 480 m s$^{-1}$. In general, the velocity field of S2 contains velocity values that range between 180 m s$^{-1}$ and 550 m s$^{-1}$ (Fig. 10b, S2). In the first 100 ms TWT the interval velocity increases from 180 m s$^{-1}$ to 300 m s$^{-1}$. A low-velocity zone is visible from 20 m to 80 m distance and between 100 to 400 ms TWT in the southern part of the profile with the lowest velocity value of ca. 180 m s$^{-1}$. The velocity field of S3 shows increasing shear wave velocities from ca. 160 m s$^{-1}$ to 280 m s$^{-1}$ between 0 and 100 ms TWT (Fig. 10b, S3). Similar to the

sections S1 and S2, S3 shows a low-velocity zone between ca. 200 and 350 ms TWT and 40 m to 78 m distance with the lowest value of ca. 185 m s$^{-1}$. Below this zone, the velocities almost constantly increase with depth up to ca. 555 m s$^{-1}$. The interval velocities of S4 range from ca. 180 m s$^{-1}$ to 580 m s$^{-1}$ and show no remarkable low-velocity zone (Fig. 10b, S4).

## 5.1 Interpretation

For geological interpretation the reflection patterns and the shear wave velocities were correlated with lithologies derived

from nearby boreholes (Fig. 11). Since the three boreholes located around the leaning church tower all show almost identical lithologies, only the lithologies of the research borehole Ky 1/2014 are described in detail (Fig. 11a). The first 3 m consist of anthropogenic deposits e.g. back fill, sand, gravel and debris. Clay and solifluction soils of the Quaternary are found between ca. 3 m and 7 m depth, followed by deposits of the Zechstein, in this case the Staßfurt and the Leine Formations, which are discontinuous due to several cavities and cavity fillings that indicate karst. The Staßfurt anhydrite is found between ca. 7 m

and 74 m depth, and the Stinkschiefer that belongs to the Staßfurt carbonate, reaches down to a depth of ca. 80 m. The Werra Formation at depths of 80 m to 163.8 m consists of anhydrite, carbonate and clay. The oldest Zechstein deposits are conglomerates, which are found between 163.8 and 165.2 m depth. From ca. 165.2 m to 347.7 m deposits of the Kyffhäuser Formation with sandstones, silts, argillites and conglomerates are found. At the base of this formation, the KSM Fault is detected with a 40° dip. Below, a second package of the Staßfurt- and the Werra Formations was drilled, which indicates that the KSM Fault

is a thrust fault that was probably active during the Tertiary. The KSM Fault is assumed to act as the main pathway for water to leach the Permian deposits in the near-surface.

Combining all structural and lithologic information leads to the following geological interpretation (Fig. 11b, c). The uppermost 10 to 15 m depth of the four profiles, with increasing $V_{INT}$ values of 160 m s$^{-1}$ to 300 m s$^{-1}$, are typical of unconsolidated Quaternary deposits, if compared to studies in other research areas (Gomberg et al., 2003; Brückl et al., 2005). The reflec-

tors below, down to 150 m depth, represent various Zechstein deposits. The discontinuous reflection patterns and the small offsets are indicators of faults and a strongly fractured strata. This is visible in sections S1 and S2 at ca. 40 m and 80 m depth (Fig. 10a, S1 & S2). The faults in the near-surface partly form conjugate normal faults, and the layers are displaced along these structures. The sediments are falling into these secondary openings due to the constant subrosion. As a result, depressions




are forming at the near-surface, also visible as dipping reflectors. The depressions consist of anthropogenic and Quaternary deposits and of the Staßfurt anhydrite.

The low-velocity zones occur predominantly in the proximity of these depressions at ca. 20 to 40 m depth (Fig. 10b). The depression in the W-E trending profile S1 is located ca. 8 m north of the church (Fig. 11c, S1). It has an extent of ca. 30 m and reaches a depth of ca. 20 m. The immediate proximity of this depression to the leaning tower and the diffractions at depths of 40 m, which are probably induced by small subsurface structures due to the strongly fractured underground, indicate that these subrosion structures are the main reason for the tilting of the tower. The same holds for the northern depression in the N-S trending profile S2, which is located 5 m east of the tower (Fig. 11c, S2). It shows an extent of ca. 40 m and a depth of ca. 20 m. The southern depression of S2 is located below the Schwedengasse road, 25 m south of the church, with an extent of 20 m and a depth of ca. 15 m. The depression-like structure, as imaged by profile S2 at the tower, is probably the same as the one in section S1, but viewed from another direction. The diffractions in both sections imply small-scale structures in the subsurface at depths of 40 m and below e.g. a fractured underground/strata, faults or cavities; the latter was drilled by the research boreholes. In section S3, there is indication of another depression structure (Fig. 11b, c, S3). It is located 15 m south of the church with an extent of ca. 30 m and a depth of ca. 20 m.

Section S4 shows a different image of the subsurface compared to the other seismic sections (Fig. 11c, S4). The section does not show depression structures in the near-surface, and low-velocity zones or large diffractions are also missing. Instead the main features of S4 are southward-dipping reflectors that represent a listric normal fault.

## 6   Discussion

Carrying out seismic surveys in an urban environment is always a challenge due to traffic passing, noise from construction sites and any other type of industrial noise. These factors lead to many interruptions of the surveys in Bad Frankenhausen. Two ways to reduce these disturbances would be road closure or night surveys. Due to the average profile length of only 8 m and sufficient time schedule these options were not used.

Another problem are densely built-up areas with limited space, like the medieval center of Bad Frankenhausen. In addition, the ground conditions differ from unpaved soil to paved soil to streets consisting of asphalt, concrete or cobblestone. The equipment used was accordingly adjusted to meet these special requirements.

Different seismic sources are generally available, e.g. explosive charges, hammer blows or weight drops, but these are not applicable in Bad Frankenhausen due to the lack of space and/or the resulting damage of the surface (Drijkoningen, 2003). For that reason the micro-vibrator ELVIS, which was developed at LIAG, was used for the seismic surveys (Fig. 6). Another advantage of the vibroseis technique is that, compared to impulse sources, the emitted high-precision frequency-modulated signal generates a repeatable and consistent wavelet (Klauder et al., 1960; Goupillaud, 1976; Nijhof et al., 1998; Drijkoningen, 2003).

The data processing here was based on general processing procedures as described by Krawczyk et al. (e.g. 2012) and Pugin et al. (2013a). Additionally, spectrum balancing was applied to improve the vertical resolution. Velocity analysis was



a difficult task, because reflection hyperbolas were barely visible, due to the discontinuous reflectors and weak reflection patterns (Fig. 10a). Additionally, an often observed decrease of the interval shear-wave velocities in the shallow subsurface in the uppermost 1000 ms TWT indicated low-velocity zones resulting in lateral and vertical velocity variations of an average value of $100\,\mathrm{m\,s^{-1}}$ (Fig. 10b). These velocity variations and the low-velocity zones also impaired the depth conversion process.

To prevent imaging artifacts, simplified 1D velocity profiles derived from the 2D velocity fields were used.

The discontinuous reflection patterns, the partly low amplitudes and the low-velocity zones indicate the formation of fractures, faults (Fig. 11c), and probably cavities leading to lower density and higher porosity values. The formation of these structures is mainly caused by leaching processes in the subsurface. The voids or the disturbances of the wavefield caused by the voids and other small structures are detectable by e.g., the presence of diffractions within the unmigrated sections (Fig. 10a),

if the size of the scatterer is smaller than the wavelength of the seismic signal. The voids also have an effect on the surrounding material, like the formation of fractures due to instabilities. This change of rock integrity is detectable by the presence of discontinuous reflectors and shear wave velocity variations.

Seismic velocities strongly vary depending on the physical rock properties. Shear wave velocities of anhydrite range from $2000\,\mathrm{m\,s^{-1}}$ to $3600\,\mathrm{m\,s^{-1}}$, and gypsum ranges from $700\,\mathrm{m\,s^{-1}}$ to $2700\,\mathrm{m\,s^{-1}}$, as derived from laboratory experiments (Brückl

et al., 2005; Mielsch & Priegnitz, 2012). Wave propagation within an intact rock is faster than a rock with fractures, faults and cavities, due to their disturbing effect on the wavefield (Barton, 2007). The observed velocities of the four seismic sections here, which contain a mixture of anhydrite and gypsum, are slower with a maximum value of ca. $580\,\mathrm{m\,s^{-1}}$ (Fig. 10b), because the strata is strongly fractured.

Fractures range in size from microcracks (up to a few micrometers) to faults (up to hundreds of km in length and tens of km

in depth), and there are always smaller fractures present than those imaged in the seismic sections. The horizontal resolution is determined by the radius of the Fresnel zone, the smaller the zone the higher the resolution. The radius of the Fresnel zone is determined by the wavelength, which depends on the central frequency and the velocity of the seismic wave. As a result, a high frequency and a slow seismic velocity will lead to a smaller wavelength and therefore to an improved resolution (Brückl et al., 2005). The strongly fractured strata/underground below Bad Frankenhausen and the unconsolidated sediments at the

near-surface result in slow seismic velocities between 160 and $580\,\mathrm{m\,s^{-1}}$. For the sweep signal a frequency bandwidth from 20 to 160 Hz was chosen and data analyses revealed that the useful signal mainly contains frequencies between 20 and 85 Hz. This results in high-resolution imaging of the shallow subsurface with a resolution of less than 1 m in depths down to ca. 15 m and a resolution of ca. 2–3 m in 50 m depth.

Boadu & Long (1996) found that seismic amplitudes and phases change at fractures, depending on the fracture parameters,

such as fracture length and fracture spacing. Only part of the energy is reflected at the fault itself and the rest is either scattered or guided by the fracture. This results in a loss of reflected energy and a reduced amplitude size; this effect is less pronounced for lower frequencies (L'Heureux et al., 2009). This is the case here (Fig. 10), where many small structural features induce scattering.

Depressions are visible in the seismic sections S1, S2 and S3 (Fig. 11). The tilting of the church tower and the subsidence of

the surface are associated with the subrosion structures seen in S1 and S2 (Fig. 10, 11). Seismic section S2 shows a depression



structure below the Schwedengasse. Additionally, in a small area between the church and the Schwedengasse road, ca. 10 m north of the eastern end of profile S3, slight subsidence of the surface is visible. This is also the case in the eastern part of section S3. Probably a new depression structure is forming at this location and is proceeding to the surface. For further observation of the evolution of this structure repeated LiDAR scans of this area are planned. Such sagging of layers and upward migration of

voids were also shown by Miller & Millahn (2006) and McDonnell et al. (2007) from research areas in Texas and Kansas in the USA.

To interpret the structures imaged by profile S4, the results of outcrop investigations, geological mapping (Schriel & Bülow, 1926a, b), and research and exploration boreholes were taken into consideration (Figs. 1, 11). These geological investigations indicate a west-to-east striking northward-dipping thrust fault at the southern margin of the Kyffhäuser, also known as the

Kyffhäuser-Southern-Margin Fault (KSM Fault). The Zechstein deposits of the Werra-, Staßfurt- and Leine Formations have been thrusted along this fault, which is probably an important pathway for water to leach the Zechstein deposits. Similar observations of leaching processes were made by Abelson et al. (2003) and Frumkin et al. (2011) along the western Dead Sea shoreline, where a rift valley that features several normal faults was identified as a possible water pathway. The imaging of the KSM Fault at depth of ca. 347 m was not possible because of the limited investigation depth of the used equipment and

the survey configuration. Instead another, so far unknown fault is imaged on profile S4 and also by a P-wave profile carried out along the same track. The visible structure is a southward-dipping listric normal fault and it can be observed down to a depth of ca. 120 m in the shear wave section (Figs. 10, 11) and down to ca. 350 m depth in the P wave section (Wadas et al., 2016).

The observations in Bad Frankenhausen can be compared with the results of other studies, which have determined seismic velocities in the context of characterizing subrosion-induced structures, but with a lower resolution. P-wave refraction seismics

has been used to detect the subsurface velocity structure, and the inversion of surface waves to calculate shear wave velocities. In the refraction seismics strong lateral variations of the velocities and zones with decreased seismic velocities are shown that are associated with salt dissolution (e.g., Karaman & Karadayilar, 2004; Dobecki & Upchurch, 2006). With a resolution of 15 to 20 % of the investigated depth (Briaud, 2013), this technique has a lower resolution with respect to the shear wave reflection method. Especially shear wave velocities are of importance, because of their relationship to the shear modulus, which is an

indication of the stiffness of subsurface rocks. Therefore low-velocity zones could be used to determine hazard-prone areas and this will be further investigated by future work.

The inversion of Rayleigh surface waves helps to deliver shear wave velocities (e.g. using the Multichannel Analysis of Surface Waves (MASW)). Some studies (Debeglia et al., 2006; Dobecki, 2010) revealed vertical and in particular lateral variations in the subsurface with low-velocity zones associated with loosening of shallow sediment layers during the formation

of voids and the development of sinkholes. This correlates with the results of this study (Fig. 10). The advantages of the MASW method are less acquisition and processing time compared to active shear wave reflection seismics. The disadvantages of MASW are the lower penetration depth, the inferior vertical and lateral resolutions (Ismail et al., 2014), and that the method is based on a 1D inversion process leading to smearing effects in 2D (Park & Taylor, 2010). If only an overview of the velocities is required Rayleigh surface wave inversion is a valuable tool to estimate the ground stability. For a detailed analysis the use of

shear wave reflection seismics is recommended. As shown, even in urban areas this method delivers a high-resolution image



of both the structures and the shear wave velocities in the near-surface (Polom et al., 2010; Krawczyk et al., 2012; Pugin et al., 2013b). In this study, we provide images down to a depth of ca. 100 m with a resolution of less than 1 m in the shallow subsurface.

## 7   Summary and Outlook

In this study we show the potential of shear-wave reflection seismics to identify subrosion-induced structures in the near-surface in an urban environment. The survey carried out in the medieval city center of Bad Frankenhausen confirms the benefits of this technique, e.g. to detect and characterize unstable zones. The seismic sections reveal five main features associated with subrosion:

1. laterally and vertically variable reflection patterns caused by strongly heterogeneous strata,

2. discontinuous reflectors, small-scale offsets and small normal faults, which were caused by a strongly fractured strata due to leaching of the Zechstein formations,

3. dipping reflectors and depression structures at the near-surface caused by deposits slowly sagging into secondary openings and cavities,

4. diffractions in the unmigrated seismic sections are possible indicators of cavities in the subsurface, which developed due to the subrosion processes, and,

5. laterally and vertically varying seismic velocities and low-velocity zones in the near-surface caused presumably by fractures and upward migrating cavities.

A previously undiscovered southward-dipping listric normal fault was found northward of the church. This fault probably serves as a main pathway for water to leach the Zechstein Formations below the church leading to the tilting of the tower.

Future work will comprise advanced data processing such as pre-stack migration, and additional shear wave reflection seismic surveys in Bad Frankenhausen. In order to compare the results from Bad Frankenhausen, seismic surveys in Schmalkalden in southern Thuringia, another area affected by subrosion, are planned.

## 8   Data availability

The seismic data, which was used in this paper, is the property of LIAG. The data is available from the authors upon request. Please contact the first author (Sonja Wadas) for details.

*Acknowledgements.* We would like to thank LIAG's seismic data acquisition and technical development team; Jan Bayerle, Eckhardt Groß-mann and Sven Wedig. Lutz Katzschmann from the Thuringian State Institute for Environment and Geology encouraged this work. The citizens of Bad Frankenhausen are thanked for their cooperation.



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



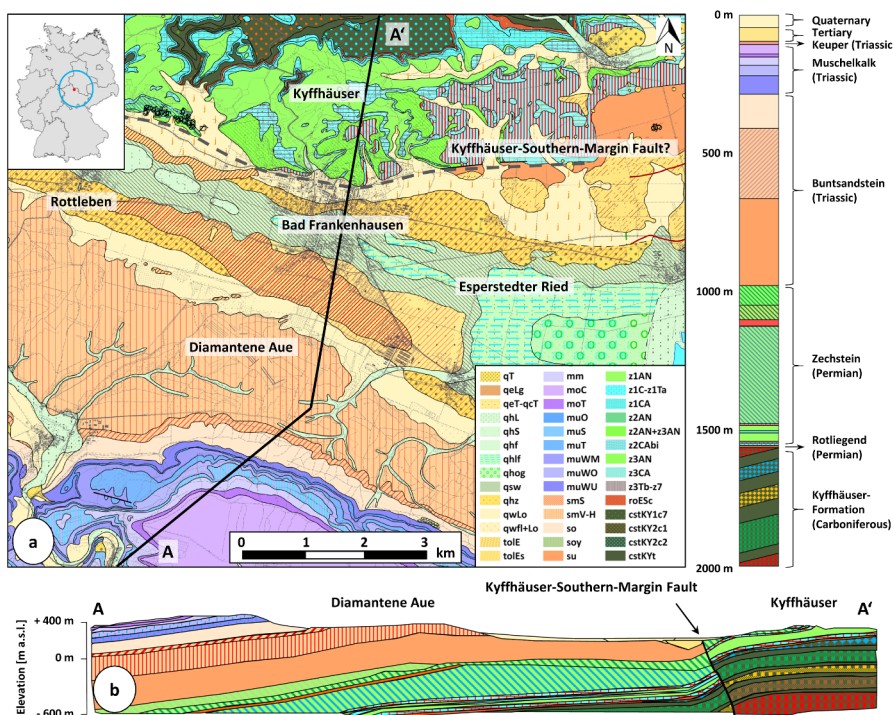

**Figure 1.** Geological setting of the research area. Bad Frankenhausen (red dot) in Thuringia (blue circle) is located centrally in Germany (see inset). (a) Geological map showing the Permian deposits of the southern part of the Kyffhäuser and the Diamantene Aue, which consists of mainly Quaternary deposits (for an explanation of the stratigraphic abbreviations see LBEG (2015). The average thickness of the geological units in the area of Bad Frankenhausen is shown right of the geological map. (b) The cross section A–A' (black line in geological map) passes through the research area and shows the northward-dipping Kyffhäuser-Southern-Margin Fault (dashed line) (after Schriel & Bülow (1926a, b)).





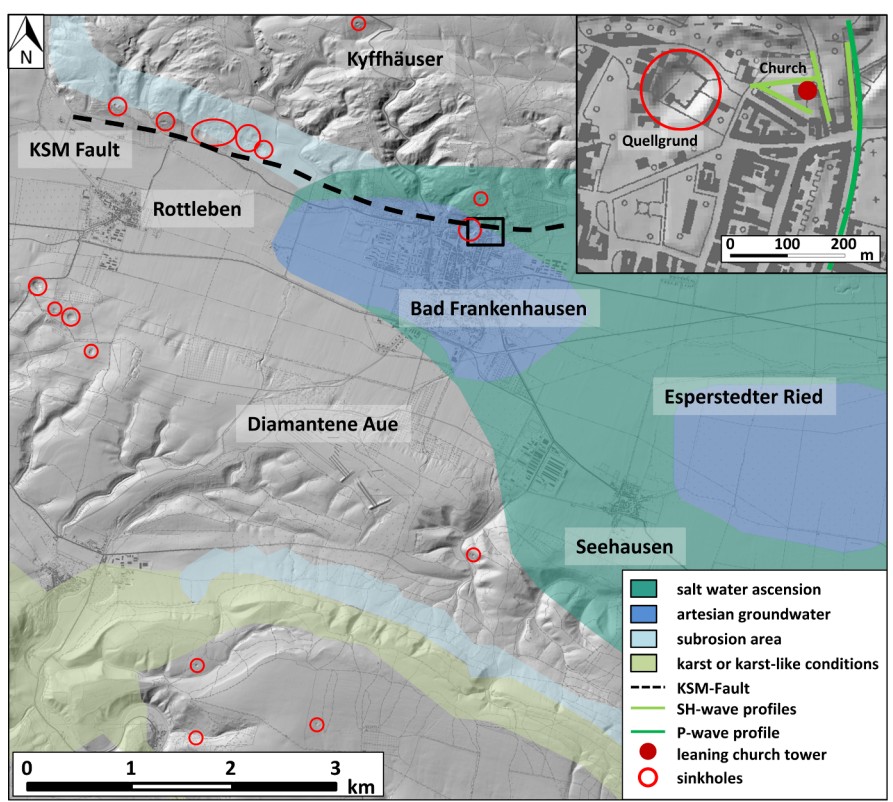

**Figure 2.** Digital elevation model showing the hydrogeological assessment of Bad Frankenhausen and the surrounding areas (provided by Thuringian State Institute of Environment and Geology, 2016). The city of Bad Frankenhausen is located in a region of salt water ascension and artesian groundwater conditions along the Kyffhäuser-Southern-Margin Fault (dashed line). The research area with the leaning church tower of the Oberkirche (red dot) and the shear wave reflection seismic profiles (yellow lines) are shown detailed in the inset. A zone influenced by subrosion is defined based on the occurrence of a line of sinkholes (red circles).



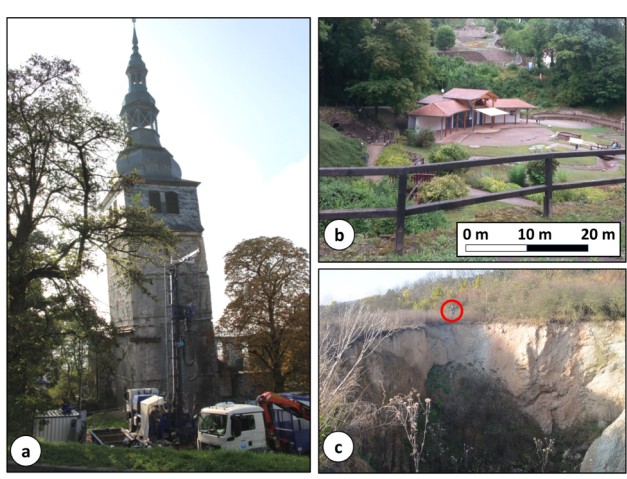

**Figure 3.** Subrosion signatures at surface in Bad Frankenhausen and the surrounding area. (a) The most famous subrosion phenomenon is the leaning church tower of Bad Frankenhausen (height of 56 m); note drilling in foreground. (b) The oldest sinkhole of the region, called Quellgrund, is located in the medieval city center of Bad Frankenhausen and used today as recreation area. (c) One of the most recent sinkholes is found directly beside the Äbtissinnen Grube (red circle marks a person standing at the sinkhole margin).




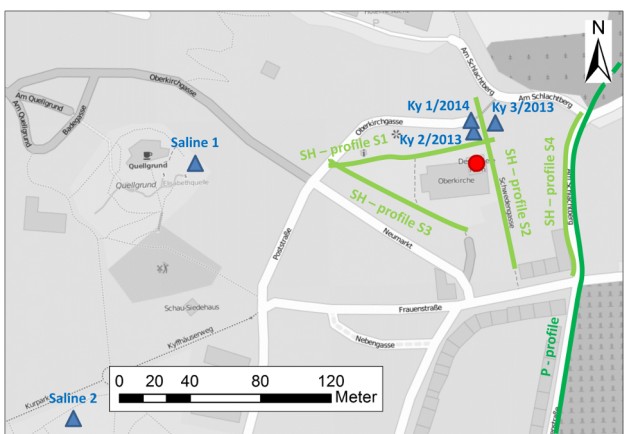

**Figure 4.** Reflection seismic profile locations in the survey area. The four SH-wave reflection seismic profiles (light green lines) were carried out around the leaning church tower of the Oberkirche (red dot) to investigate subrosion structures responsible for the constant sagging of the surface. Profile S4 runs partly along the same location as a previously carried out P-wave reflection seismic profile (dark green line), enabling the comparison of both seismic sections. Close to the Oberkirche 5 boreholes exist for calibration (blue triangles; for a detailed description see Fig. 11). Saline 1 and Saline 2 are historical boreholes from 1857 and 1866, drilled to extract brine. Their stratigraphy is rather approximate. Ky 2/2013 and Ky 3/2013 are pre-investigation boreholes for the deep research borehole Ky 1/2014. These wells are well documented, including e.g. stratigraphy, density, susceptibility, gamma ray logging, temperature and conductivity.




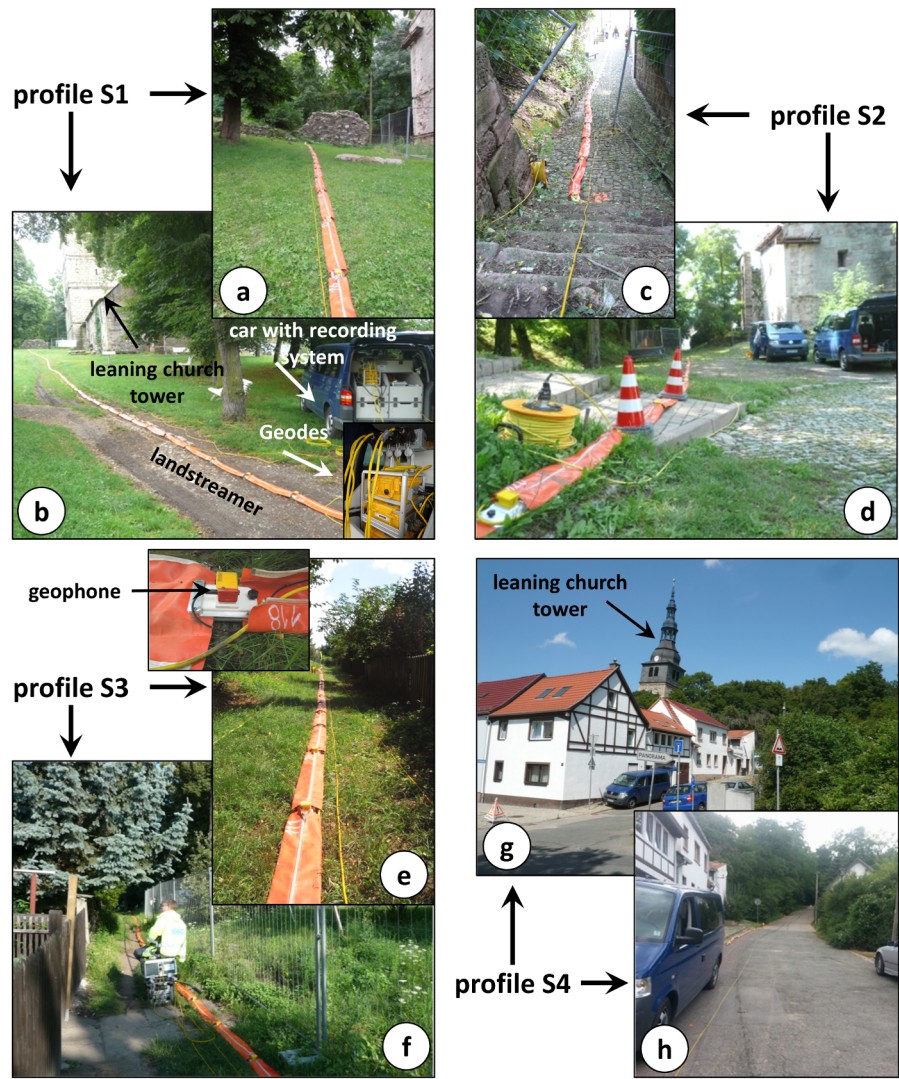

**Figure 5.** Field campaign in July 2014 under variable survey conditions. The shear-wave reflection seismic profile S1 (a) and (b) was carried out next to the church with the landstreamer (orange covering) crossing over remains of walls of the medieval city. Profile S2 (c) and (d) runs east of the leaning church tower. An additional challenge are stairs along the profile track. Profile S3 (e) and (f) was carried out behind the church, adjacent to private gardens on a ca. 0.5 m narrow path. Profile S4 (g) and (h) was carried out along the street Am Schlachtberg.




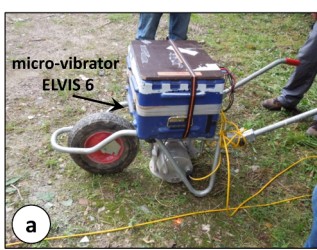
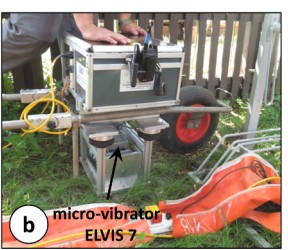

**Figure 6.** Field campaign in July 2014 using different seismic sources. (a) Micro-vibrator ELVIS 6 on paved ground, (b) micro-vibrator ELVIS 7 on unpaved ground and landstreamer beside it. Both vibrators are ideally suited for application in densely built-up areas such as the medieval city center of Bad Frankenhausen.



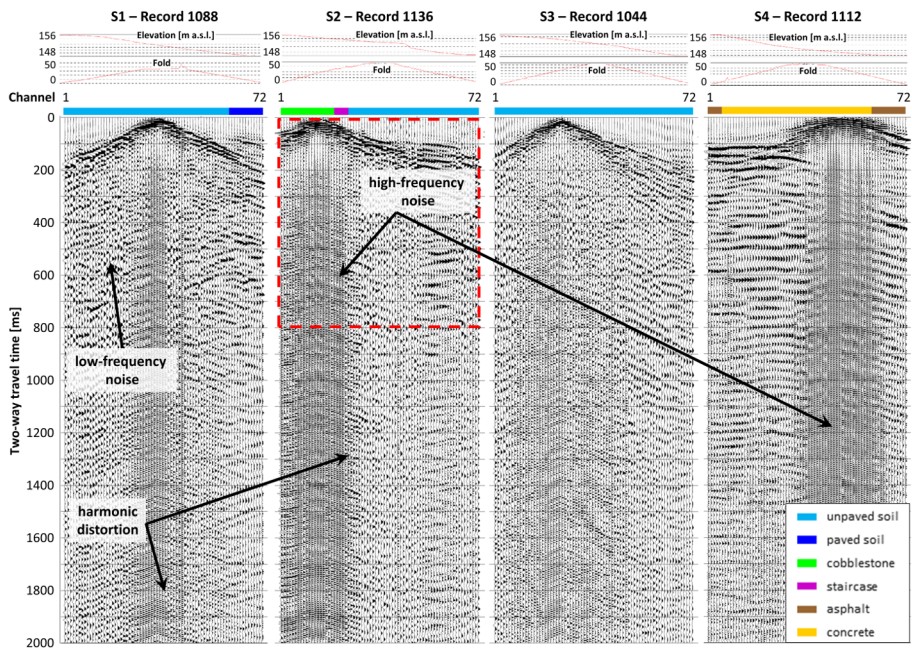

**Figure 7.** Comparison of records of the four SH-wave seismic profiles showing the differing data quality although the four profiles are not far from each other (c.f. Fig. 4). The main controlling factor are near source inhomogeneities in the subsurface leading to harmonic distortions, which cover the reflection signals by high-frequency noise. Also the coupling quality on unpaved soil or cobblestone is impaired compared to a paved surface e.g. an asphalt- or concrete-paved road. Besides the harmonic distortions, low- and high-frequency noise caused by environmental noise also drown the reflection signals, especially on profiles S1, S2, and S3.



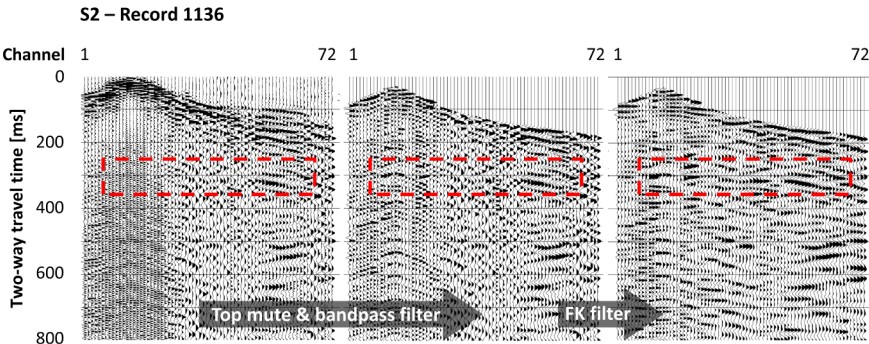

**Figure 8.** To eliminate noise and harmonic distortions, a bandpass filter and individual FK filters were applied to the records of each profile. The filtering in the frequency-wave number domain was one of the key processing steps to get interpretable seismic sections (note red box, showing the improved S/N-ratio of near-surface reflectors).



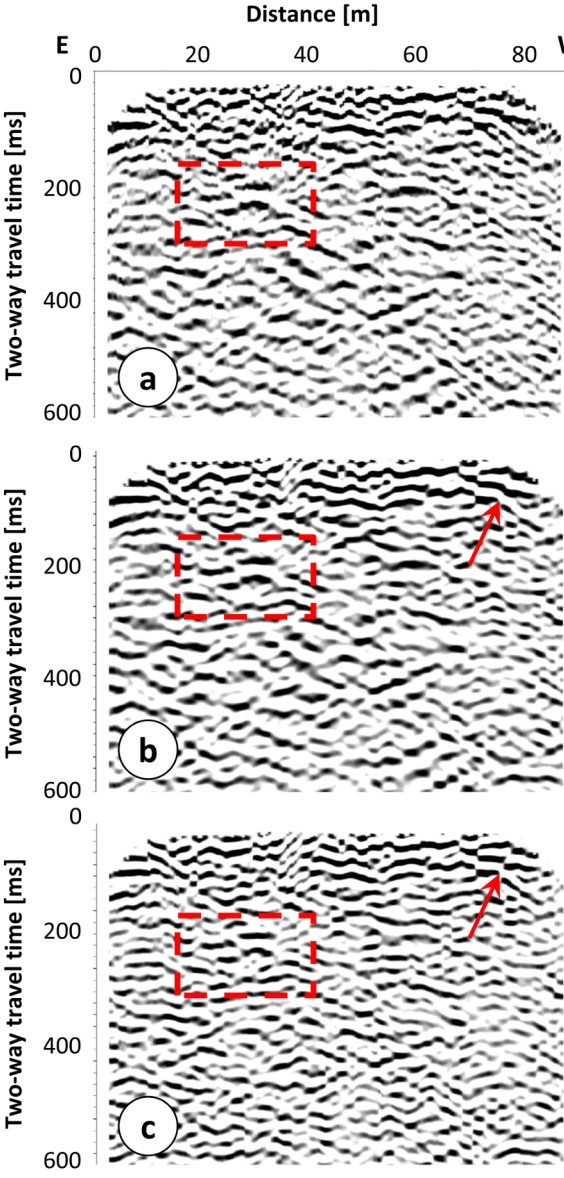

**Figure 9.** CMP-stacked sections of profile S1 after different processing steps. (a) The first CMP-stacked section shows residual noise. (b) After applying a bandpass filter to the stacked data the reflectors have an improved S/N ratio, especially from 0 to 400 ms TWT. (c) By using spectral balancing the lateral resolution is further improved (note red boxes and arrows showing the improved resolution of near-surface reflectors).







**Figure 10.** Shear-wave reflection seismic profiles S1–S4 with (a) stack in time domain showing diffractions and color-coded amplitudes and (b) FD migration in time domain with color-coded interval velocities. Position of the church tower is marked and blue arrows represent intersection points of profiles.





**Figure 11.** Five boreholes, located in the proximity of the church, give information about the lithologies (a). Since the three boreholes located around the leaning church tower all show almost identical lithologies, only the lithologies of the research borehole Ky 1/2014 were used for correlation of the reflectors in the migrated and depth converted seismic sections S1–S4 (b). The result are four interpreted sections (c). The dashed lines show the internal stratification of the Quaternary and the Staßfurt anhydrite. The church tower is marked and the blue arrows on top mark intersection points of profiles.



**Table 1.** Device and acquisition parameters of the seismic source, the receivers, and the recording system.

|  | Device specifications | Acquisition parameters |
|---|---|---|
| **Source** | micro-vibrators ELVIS 6 and 7 (weight 95 kg, peak force 1 kN, frequency range for shear waves 20–250 Hz) | sweep frequency 20–160 Hz, sweep duration 10 s source spacing 2 m, 4 excitations/point |
| **Receiver** | horizontal geophones (Sensor, type SM 6, resonance frequency 10 Hz, electric resistance 375 Ohm (coil) & 1000 Ohm (damping)) | 72 geophones mounted on a landstreamer, spacing 1 m, fixed configuration, mean CMP-fold 18 |
| **Recording system** | Geode (Geometrics Inc.) | 73 channels (72 data channels + 1 pilot sweep channel), record length 12 s (2 s correlated), sampling interval 1 ms |





**Table 2.** Overview of the general processing sequence applied to the shear-wave reflection seismic data. Most of the processes were carried out iteratively. They are all individualized for differing data quality of the four profiles.

| Processing step | Parameter |
|---|---|
| Geometry check | Manual correction of geodetic data to eliminate errors caused by DGPS reference points (mean errors are 5 cm for S1, 8 cm for S2, 2 cm for S3 and 5 cm for S4) |
| Quality control | Examination of seismic data |
| Vibroseis correlation | Cross-correlation of the sweep (frequency 20–160 Hz, length 10 s) and the recorded signal (length 12 s) |
| Geometry | Combination of GPS data and seismic data, crooked line binning using 0.5 m bin interval |
| Amplitude & spectral editing | Automatic Gain Control using a window of 200 ms length, a Bandpass Filter (15/17 Hz–163/165 Hz pass the filter) & an Amplitude normalization |
| Vertical stack | 4-fold |
| Top mute | Individual zeroing of amplitudes at the top of the records |
| FK filter | Filtering out surface waves and other coherent noise using individual polygon filters in the frequency-wavenumber domain (e.g., FK filter of S1 on the right) |
| CMP sort | Sort from shot gathers to CMP gathers |
| Interactive velocity analysis | Analysis interval was 5 m to 10 m using 11 or 12 CMPs per analysis location |
| NMO correction | Shift hyperbolas to zero-offset traveltimes |
| Residual statics correction | Correction of near-surface velocity variations |
| CMP stack | Mean fold is 18 traces per CMP |
| Filter | Filtering out the remaining noise using an individual Bandpass Filter for each profile (e.g., 17/19 Hz–73/75 Hz) |
| Spectrum balancing | Individual correction of frequency attenuation to improve the resolution (e.g., bandwidth 15 Hz, slope 5 Hz, start 20 Hz, end 70 Hz) |
| FD migration | Movement of dipping reflectors, collapse of diffractions and increase in spatial resolution (filtered 45–65 degree) |
| Depth conversion | Conversion from time to depth |