# Peer review of "High-resolution shear wave reflection seismics as tool to image near-surface subrosion structures — a case study in Bad Frankenhausen, Germany"

_Solid Earth, 2016_

## Referee Comment (RC1) · Anonymous Referee #1 · 9 Aug 2016

I have read your paper with great interest. You present a case study where you use shear wave reflection seismic to image near-surface subrosion structures. New data are presented resulting in a subsurface model that could explain the inclination of the church tower. The manuscript is well organized and the authors followed the journal guidelines. The used methodology is sound, valid and clearly outlined. The analysis and findings support the type of publication (case study) in fields (geophysics, structural geology, and tectonics) that are in the scope of SE.

Overall it is an interesting, well elaborated and solid case study. Therefore I recommend

the manuscript for publication with minor revision.

The following comments are suggestions and I hope you find them useful in improving the quality of your manuscript:

Specific Comments:

While it is interesting and might for some readers be appealing, some parts of the manuscript are written in great detail and a bit repetitive. The manuscript could gain much from more conciseness.

The manuscript seems a bit over-referenced, especially the introduction and geological setting chapters are highly fragmented by numerous references, some of them repeated several times. Further to that more than 20% of the references are in German which for an international audience is of no use.

I missed an explanation how the velocity analysis was done. That doesn't need to be a big paragraph but maybe just stating the type of analysis that was used to gain the velocity field.

For critique on the figures please refer to the next paragraph.

Technical Corrections:

Please be careful with the use of 'seismics' as opposed to seismic. 'Seismics' it is not a noun and often only used in the informal language or colloquial speech (probably check with the desk editor in which way it is used in the journal).

Ulriksen in text – Ullriksen in bibliography

'Knoth & Schwab, 1972' in bibliography but not in text, same for 'Polom, 2013'

Figures: A few figures suffer from small annotation. While it is fine to have them on screen, for the printed journal they seem to be too small (Figs 1, 2, 4 and 7, please also see annotated manuscript).

Figure 10 and 11: A seismic interested person would certainly like to see the seismic data in a larger display. Both figures could gain from: - not showing all data down to 1200ms or 150m - omitting some of the repeated time and depth axis and annotations - less gaps between the panels

Figure 11: I personally think that the interpreted results should be overlaid on the seismic. I find it very hard to transfer the faults and especially the depressions from 11c to 11b and therefore hard to follow the interpretation at all.

For a few others suggestions please see also annotated manuscript.

Please also note the supplement to this comment:
http://www.solid-earth-discuss.net/se-2016-91/se-2016-91-RC1-supplement.pdf

**Supplement:**

[revised manuscript text omitted]

*annotation for printed version too small, especially street names which are referred to in text*

**Figure 4.** Reflection seismic profile locations in the survey area. The four SH-wave reflection seismic profiles (light green lines) were carried out around the leaning church tower of the Oberkirche (red dot) to investigate subrosion structures responsible for the constant sagging of the surface. Profile S4 runs partly along the same location as a previously carried out P-wave reflection seismic profile (dark green line), enabling the comparison of both seismic sections. Close to the Oberkirche 5 boreholes exist for calibration (blue triangles; for a detailed description see Fig. 11). Saline 1 and Saline 2 are historical boreholes from 1857 and 1866, drilled to extract brine. Their stratigraphy is rather approximate. Ky 2/2013 and Ky 3/2013 are pre-investigation boreholes for the deep research borehole Ky 1/2014. These wells are well documented, including e.g. stratigraphy, density, susceptibility, gamma ray logging, temperature and conductivity.

[Figure]

[Figure]

**Figure 5.** Field campaign in July 2014 under variable survey conditions. The shear-wave reflection seismic profile S1 (a) and (b) was carried out next to the church with the landstreamer (orange covering) crossing over remains of walls of the medieval city. Profile S2 (c) and (d) runs east of the leaning church tower. An additional challenge are stairs along the profile track. Profile S3 (e) and (f) was carried out behind the church, adjacent to private gardens on a ca. 0.5 m narrow path. Profile S4 (g) and (h) was carried out along the street Am Schlachtberg.

[Figure]

[Figure]

[Figure]

**Figure 6.** Field campaign in July 2014 using different seismic sources. (a) Micro-vibrator ELVIS 6 on paved ground, (b) micro-vibrator ELVIS 7 on unpaved ground and landstreamer beside it. Both vibrators are ideally suited for application in densely built-up areas such as the medieval city center of Bad Frankenhausen.

[Figure]

[Figure]

[Figure]

**Figure 7.** Comparison of records of the four SH-wave seismic profiles showing the differing data quality although the four profiles are not far from each other (c.f. Fig. 4). The main controlling factor are near source inhomogeneities in the subsurface leading to harmonic distortions, which cover the reflection signals by high-frequency noise. Also the coupling quality on unpaved soil or cobblestone is impaired compared to a paved surface e.g. an asphalt- or concrete-paved road. Besides the harmonic distortions, low- and high-frequency noise caused by environmental noise also drown the reflection signals, especially on profiles S1, S2, and S3.

[Figure]

[Figure]

**Figure 8.** To eliminate noise and harmonic distortions, a bandpass filter and individual FK filters were applied to the records of each profile. The filtering in the frequency-wave number domain was one of the key processing steps to get interpretable seismic sections (note red box, showing the improved S/N-ratio of near-surface reflectors).

*this figure is certainly worth being shown as big as figure 9, maybe in the same manner*

[Figure]

[Figure]

[Figure]

**Figure 9.** CMP-stacked sections of profile S1 after different processing steps. (a) The first CMP-stacked section shows residual noise. (b) After applying a bandpass filter to the stacked data the reflectors have an improved S/N ratio, especially from 0 to 400 ms TWT. (c) By using spectral balancing the lateral resolution is further improved (note red boxes and arrows showing the improved resolution of near-surface reflectors).

*Suggestion: Show in same manner as Figure 8 with arrows indicating processing steps*

[Figure]

**Figure 10.** Shear-wave reflection seismic profiles S1–S4 with (a) stack in time domain showing diffractions and color-coded amplitudes and (b) FD migration in time domain with color-coded interval velocities. Position of the church tower is marked and blue arrows represent intersection points of profiles.

[Figure]

**Figure 11.** Five boreholes, located in the proximity of the church, give information about the lithologies (a). Since the three boreholes located around the leaning church tower all show almost almost identical lithologies, only the lithologies of the research borehole Ky 1/2014 were used for correlation of the reflectors in the migrated and depth converted seismic sections S1–S4 (b). The result are four interpreted sections (c). The dashed lines show the internal stratification of the Quaternary and the Staßfurt anhydrite. The church tower is marked and the blue arrows on top mark intersection points of profiles.

[Figure]

**Table 1.** Device and acquisition parameters of the seismic source, the receivers, and the recording system.

| | Device specifications | Acquisition parameters |
|---|---|---|
| **Source** | micro-vibrators ELVIS 6 and 7 (weight 95 kg, peak force 1 kN, frequency range for shear waves 20–250 Hz) | sweep frequency 20–160 Hz, sweep duration 10 s source spacing 2 m, 4 excitations/point |
| **Receiver** | horizontal geophones (Sensor, type SM 6, resonance frequency 10 Hz, electric resistance 375 Ohm (coil) & 1000 Ohm (damping)) | 72 geophones mounted on a landstreamer, spacing 1 m, fixed configuration, mean CMP-fold 18 |
| **Recording system** | Geode (Geometrics Inc.) | 73 channels (72 data channels + 1 pilot sweep channel), record length 12 s (2 s correlated), sampling interval 1 ms |

[Figure]

**Table 2.** Overview of the general processing sequence applied to the shear-wave reflection seismic data. Most of the processes were carried out iteratively. They are all individualized for differing data quality of the four profiles.

| Processing step | Parameter |
| --- | --- |
| Geometry check | Manual correction of geodetic data to eliminate errors caused by DGPS reference points (mean errors are 5 cm for S1, 8 cm for S2, 2 cm for S3 and 5 cm for S4) |
| Quality control | Examination of seismic data |
| Vibroseis correlation | Cross-correlation of the sweep (frequency 20–160 Hz, length 10 s) and the recorded signal (length 12 s) |
| Geometry | Combination of GPS data and seismic data, crooked line binning using 0.5 m bin interval |
| Amplitude & spectral editing | Automatic Gain Control using a window of 200 ms length, a Bandpass Filter (15/17 Hz–163/165 Hz pass the filter) & an Amplitude normalization |
| Vertical stack | 4-fold |
| Top mute | Individual zeroing of amplitudes at the top of the records |
| FK filter | Filtering out surface waves and other coherent noise using individual polygon filters in the frequency-wavenumber domain (e.g., FK filter of S1 on the right) |
| CMP sort | Sort from shot gathers to CMP gathers |
| Interactive velocity analysis | Analysis interval was 5 m to 10 m using 11 or 12 CMPs per analysis location |
| NMO correction | Shift hyperbolas to zero-offset traveltimes |
| Residual statics correction | Correction of near-surface velocity variations |
| CMP stack | Mean fold is 18 traces per CMP |
| Filter | Filtering out the remaining noise using an individual Bandpass Filter for each profile (e.g., 17/19 Hz–73/75 Hz) |
| Spectrum balancing | Individual correction of frequency attenuation to improve the resolution (e.g., bandwidth 15 Hz, slope 5 Hz, start 20 Hz, end 70 Hz) |
| FD migration | Movement of dipping reflectors, collapse of diffractions and increase in spatial resolution (filtered 45–65 degree) |
| Depth conversion | Conversion from time to depth |

---

## Referee Comment (RC2) · J. Kammann (Referee) · 14 Sep 2016

**General Comments**

This is a very interesting case study using seismic data acquired with the ELVIS shear wave vibrator source to map subrosion structures in the urbanized area of Bad Frankenhausen. Four new high-resolution shear wave profiles are presented showing faults and fractured areas that correlate with low seismic velocities. The authors conclude that these structures are caused by subrosion in the near surface and are most probably the cause of the inclination of a church tower. The used scientific methods

and assumptions are clearly presented. Results and discussion are elaborate and sufficiently support the interpretation and conclusions of the presented study. The abstract is concise and summarizes the main points.

The article is well structured and the topic (geophysics, structural geology) is relevant for Solid Earth. I therefore recommend it for publication with minor revisions.

I have a few comments which are suggestions that I hope may help in improving the quality of the paper.

Specific Comments

The sections "Seismic survey" and "Data processing" appear partly too detailed, especially the description of pre-processing steps. Shorter and more concise description emphasizing the essential information on the measuring setup and processing would benefit the reader. However, as the FK-filter is crucial, this part could be supported with a plot of the FK-spectrum and the applied filter in figure 8. Alternatively, make a reference to the FK-plot in table 2.

Figure 4 shows the location of a P-wave profile which is further discussed in chapter 6 for comparison with profile S4. For me as a reader, it would be interesting to compare these two profiles, however, the profile was not accessible as referenced (Wadas et. al., 2016). Please ensure that the work is referenced properly or include the P-wave data.

The authors mention that the profiles were acquired with two different generations of the ELVIS source and two pictures are provided in figure 6. I am missing a comparison of these sources, what are the differences and which profiles were acquired with which source? If there are no differences in data quality, mentioning the two source types and the figure seem unnecessary.

Technical Comments

The numbering of the figures, parenthesis and use of capital letters should be checked

in the text.

There are some inconsistencies writing S/N ratio or signal-to-noise ratio in the text, please check.

Please also check that the terms "reflector" and "reflection" are used correctly.

The term Tertiary is not used anymore and should be substituted by Paleogene or Neogene (e.g. Page 3 Line 29 and following, Figure 1).

Page 3 line 12 Kelbra is not shown in the map

Page 4 line 2 . . .south of Kyffhäuser. . .

On page 4 "3 Seismic survey" the chapter starts by giving a general introduction of the position and elevation of the profiles. Adding the length of the profiles would give an idea about the special dimension of the study.

Page 4 line 14 (Fig. 2)

Page 4 line 18 (Fig. 3a)

Page 4 line 20 ". . .which exceeds the inclination of the leaning tower of Pisa at 3.97"

Page 5 line 2 the authors write about profile 4: "To meet the requirements of this challenging investigation area the equipment and the configuration used for the shear wave reflection seismics had to be adapted by splitting the streamer." – It is not clear what is meant by that as the receiver number seems to be constant for all profiles (figure 7). Please clarify what is meant.

Page 5 line 7 ". . .(e.g. Dasios et al., 1999; Inazaki, 2004)."

Page 5 line 6 The authors write: "This source-receiver combination reduces the converted waves", this however is slightly incorrect. The reduction of converted waves is achieved by vertical stacking of opposite polarization of the SH data after cross-correlation.

Page 5 line 32 "Therefore the mean CMP-fold of the profiles results in 18 traces. . ."

Page 6 line 14 ". . .improve the signal-noise ratio, by. . .)

Page 6 line 23 "Besides the vertical stack the S/N ratio was improved. . ."

Page 7 line 28 ". . .that have vertical offsets of ca. 1 m. . ."

Page 8 line 29 "reflections observed in the seismic image" instead of "reflectors"

Page 9 line 13 "Section 3 reveals another 30m wide and 20 deep depression structure (Fig. 11b, c, S3), located 15 m south of the church."

Page 9 line 16 ". . . in the near-surface, low velocity zones or significant diffractions."

Page 9 line 21 "80 m"

Page 9 line 24 ". . .to asphalt, concrete or cobblestone streets."

Page 11 line22 ". . . are shown to be associated with salt dissolution (. . .)."

Page 12 In section "Summary and Outlook" the planned LiDAR scans could be mentioned as well (or instead)

Figure 1 could be more clear and maybe simplified, especially the legend. The reference LBEG (2015) is not in the reference list. The caption is unclear. Also the borehole information used for interpretation of the shear-wave data differ a lot from the average stratification.

Figure 7 compares 4 shots of the different profiles in means of noise and surface conditions. Which processing steps were applied to these shots?

---

## Author Comment (AC1) · 28 Sep 2016

We want to thank Janina Kammann and an anonymous referee, who invested their precious time writing the reviews and therefore helped improving the manuscript.

The PDF file contains the reviewer comments, the corresponding author comments, and the differences between the two manuscript versions. Any removed words are crossed out with a single line and colored red, whereas any added words are under-lined with a squiggle and colored blue.

Modifications of images are described in in the table.

[Figure]

Please also note the supplement to this comment:
http://www.solid-earth-discuss.net/se-2016-91/se-2016-91-AC1-supplement.pdf

––––––––––––––––––––––––––––––––––

[Figure]

**Supplement:**

REVIEW 1: *Anonymous Referee* *(minor revision)*

REVIEW 2: *Janina Kammann; University of Copenhagen* *(minor revision)*

| Section | Reviewer | Note | agreed | not agreed | comment |
|---|---|---|---|---|---|
| **Abstract** | Anonymous: | *page 1 line 9-10 - word 'strongly' twice in same sentence* | X | | We replaced the second 'strongly' with 'heavily'. |
| **1 Introduction** | Anonymous: | *page 2 line 15-18 - 'can also do' = expression* | X | | We have removed the colloquial language. |
| | | *page 2 line21 - 'Ulrikse, 1982' = spelling* | X | | Spelling mistake corrected. |
| **2 Geological setting** | Anonymous: | *page 4 line 1 - '...1962)(Fi...' = missing blank space* | X | | We added the missing space. |
| | Kammann: | *page 3 line 12- Kelbra is not shown in the map* | X | | We removed the unimportant information. |
| | | *page 4 line 2 -. . .south of Kyffhäuser. . .* | X | | We changed '...south of the Kyffhäuser is ...' to 'south of the Kyffhäuser hills is ...'. |
| | | *page 4 line 14 - (Fig. 2)* | X | | Sorry, the images of figure 3 did not match the numbering within the text. We have changed the order of the images. |
| | | *page 4 line 18 - (Fig. 3a)* | X | | Sorry, the images of figure 3 did not match the numbering within the text. We have changed the order of the images. |
| | | *page 4 line 20 - '. .which exceeds the inclination of the leaning tower of Pisa at 3.97'* | X | | We changed the text in accordance with the suggestion of the reviewer. |
| **3 Seismic survey** | Kammann: | The authors mention that the profiles were acquired with two different generations of the ELVIS source and two pictures are provided in figure 6. I am missing a comparison of these sources, what are the differences and which profiles were acquired with which source? If there are no differences in data quality, mentioning the two source types and the figure seem unnecessary. | X | | Profile S1 was aquired with ELVIS 6 and S2, S3 and S4 were aquired with ELVIS 7. We now inserted this information into the text. Both sources are basically comparable, as described in the text. Comparison of single records acquired with ELVIS 6 and ELVIS 7 reveal minor differences. The older ELVIS 6 has a stronger weighting of the lower frequencies and the new ELVIS 7 has a stronger weighting of the higher frequencies, resulting in better resolution. After data processing, these minor differences are not visible anymore because of the energy loss of higher frequencies, since the useful signal of our data contains mostly frequencies below 80 Hz. However, as this article is a case study and not a technical paper, we decided not to go into too much detail regarding the electrodynamic micro-vibrators, because information about this source have already been published by, e.g. Krawczyk et al., 2012/2013 or Polom 2003/2013. Thereby, we keep the main focus on the imaging of near-surface subrosion structures. |
| | | On page 4 "3 Seismic survey" the chapter starts by giving a general introduction of the position and elevation of the profiles. Adding the length of the profiles would give an idea about the special dimension of the study. | X | | We added the length of each seismic profile. |
| | | *page 5 line 2 - the authors write about profile 4: "To meet the requirements of this challenging investigation area the equipment and the configuration used for the shear wave reflection seismics had to be adapted by splitting the streamer." – It is not clear what is meant by that as the receiver number seems to be constant for all profiles (figure 7). Please clarify what is meant.* | X | | We added 'see below for detailed description' at the end of this paragraph. The explanation is given later, when the survey setup is described in detail. Here the explanation from our text: 'Prior to survey, the landstreamer was adapted for the limited space in Bad Frankenhausen and separated into three parts for manual handling, each containing 24 geophones.' |
| | | *page 5 line 7 - '. . .(e.g. Dasios et al., 1999; Inazaki, 2004)'.* | X | | We added 'e.g.' to the reference. |
| | | *page 5 line 6 - The authors write: "This source-receiver combination reduces the converted waves", this however is slightly incorrect. The reduction of converted waves is achieved by vertical stacking of opposite polarization of the SH data after crosscorrelation.* | X | | We have explained this now more detailed for a better understanding. |
| | | *page 5 line 32 - 'Therefore the mean CMP-fold of the profiles results in 18 traces. . .'* | X | | We changed the text in accordance with the suggestion of the reviewer. |
| **4 Data processing** | Anonymous: | *page 6 line 29-60 - I missed an explanation how the velocity analysis was done. That doesn't need to be a big paragraph but maybe just stating the type of analysis that was used to gain the velocity field.* | X | | In a short sentence we now state the type of velocity analysis |
| | | *page 7 line 2 - 'although' = ?* | X | | The word 'although' was misplaced. We removed it. |
| | Kammann: | *The sections "Seismic survey" and "Data processing" appear partly too detailed, especially the description of pre-processing steps. Shorter and more concise description emphasizing the essential information on the measuring setup and processing would benefit the reader. However, as the FK-filter is crucial, this part could be supported with a plot of the FK-spectrum and the applied filter in figure 8. Alternatively, make a reference to the FK-plot in table 2.* | X | | We added a reference to the FK plot in Table 2 and shortened the text of the 'Seismic survey' section. |
| | | *page 6 line 14 - '. .improve the signal-noise ratio, by. . .)'* | X | | We changed the text in accordance with the suggestion of the reviewer. |
| | | *page 6 line 23 - 'Besides the vertical stack the S/N ratio was improved. . .'* | X | | We changed the text in accordance with the suggestion of the reviewer. |
| **5 Structural imaging and Interpretation** | Anonymous: | *page 7 line 16 - change '100-150 ms' to '100 to 150 ms'* | *X* | | We changed '-' to 'to' in the entire section. |
| | | *page 7 line18 - 'between 30 m and 80 m distance and 100 to 200 ms TWT' = hard to see in small figure* | X | | Now the images in Figure 10 are bigger. We shortened the sections down to 1000 ms instead of imaging 1200 ms, and we have omitted two of the repeated time axes. |
| | | *page 7 line 25 - insert 'but more reflectivity'* | X | | We changed the text in accordance with the suggestion of the reviewer. |
| | | *page 7 line 26 - 'similar for all profiles' = not true* | X | | Sorry for the mistake. In this part we describe the similarities of profiles S1, S2 and S3. We have changed this to make it clear. |
| | | *page 7 line 32 - change '...VINT of the seismic sections S1, S2 and S3 do not constantly increasing with depth, rather...' to '...VINT in the seismic sections S1, S2 and S3 do not constantly increase with depth, in fact...'* | X | | We changed the text in accordance with the suggestion of the reviewer. |
| | | *page 8 line 12 - describe differences, similarities as you did in structural interpretation* | X | | We added a short comparision of S4 to S1, S2 and S3. |
| | Kammann: | *page 7 line 28 - '. .that have vertical offsets of ca. 1 m. . .'* | X | | We changed the text in accordance with the suggestion of the reviewer. |
| **5.1 Interpretation** | Anonymous: | *page 8 line 32 - 'Fig. 10a' = is Fig. 11b meant here?* | X | | Yes, we meant Figure 11 b. |
| | | *page 9 line 3 - 'Fig. 10b' = is Fig. 11c meant here?* | X | | The numbering is correct. The figure reference is referring to the low-velocity zones mentioned at the beginning of the sentence. For a better understanding we have shifted the reference from the end to the beginning. |
| | Kammann: | *page 8 line 29 - 'reflections observed in the seismic image' instead of 'reflectors'* | X | | We changed 'reflectors' to 'reflections'. |
| | | *page 8 line 13 - 'Section 3 reveals another 30m wide and 20 deep depression structure (Fig. 11b, c, S3), located 15 m south of the church.'* | X | | We changed the text in accordance with the suggestion of the reviewer. |
| | | *page 9 line 16 - '. . in the near-surface, low velocity zones or significant diffractions.'* | X | | We changed the text in accordance with the suggestion of the reviewer. |
| **6 Discussion** | Anonymous: | *page 9 line 21 - '...only 8 m...' = '...only 80 m...'* | X | | Sorry for the mistake. Of course the average profile length is 80 m. |
| | | *page 10 line 8 - 'voids' = rephrase* | X | | We changed the first 'voids' to 'cavities' . |
| | | *page 11 line 15 - 'P-wave profile' = show data and faullt* | X | | The P-wave section is not shown, because the data is still in the processing stage. In the future, this P-wave seismic section, together with other P- and SH-wave profiles, will be shown in a separate publication regarding the regional geology of Bad Frankenhausen. We left out this information here, and changed Fig. 4 accordingly. |
| | | *page 11 line 34 - insert comma* | X | | We changed the text in accordance with the suggestion of the reviewer. |
| | | *page 12 line 3 - insert 'upper 15 m'* | X | | We changed the text in accordance with the suggestion of the reviewer. |
| | Kammann: | *page 9 line 21 - '80 m'* | X | | Sorry for the mistake. Of course the average profile length is 80 m. |
| | | *page 9 line 24 - '. .to asphalt, concrete or cobblestone streets.'* | X | | We changed the text in accordance with the suggestion of the reviewer. |

| # | Section | Reviewer | Comment | | Response |
|---|---|---|---|---|---|
| 36 | | | *page 11 line 15 - Figure 4 shows the location of a P-wave profile which is further discussed in chapter 6 for comparison with profile S4. For me as a reader, it would be interesting to compare these two profiles, however, the profile was not accessible as referenced (Wadas et. al., 2016). Please ensure that the work is referenced properly or include the P-wave data.* | X | The P-wave section is not shown, because the data is still in the processing stage. In the future, this P-wave seismic section, together with other P- and SH-wave profiles, will be shown in a separate publication regarding the regional geology of Bad Frankenhausen. We left out this information here, and changed Fig. 4 accordingly. |
| 37 | | | *page 11 line 22 - '. . are shown to be associated with salt dissolution (. . .).'* | X | We changed the text in accordance with the suggestion of the reviewer. |
| 38 | **7 Summary and Outlook** | Kammann: | *page 12 - In section "Summary and Outlook" the planned LiDAR scans could be mentioned as well (or instead).* | X | Now the planned LiDAR scans are mentioned in section 'Summary and Outlook'. |
| 39 | References | Anonymous: | The manuscript seems a bit over-referenced, especially the introduction and geological setting chapters are highly fragmented by numerous references, some of them repeated several times. | X | We removed some repeated or unnecessary references. |
| 40 | | | Further to that more than 20% of the references are in German which for an international audience is of no use. | X | We are aware of the large amount of German references. Since most of the data about the geological setting of our research area were acquired by German scientists a few decades ago, at a time Bad Frankenhausen was located in the German Democratic Republic, the common scientific language was German. Research in this area started to be published in English only during the 90s. However, we removed some of the repeated or unnecessary German references. |
| 41 | | | *page 16 line 6 & page 17 line 24 - 'Knoth & Schwab, 1972' in bibliography but not in text, same for 'Polom, 2013'* | X | We removed the references from the bibliography. |
| 42 | | | *page 18 line 20 - Ulriksen in text – Ullriksen in bibliography* | X | We corrected the spelling mistake. |
| 43 | Figures | Anonymous: | A few figures suffer from small annotation. While it is fine to have them on screen, for the printed journal they seem to be too small. | X | We checked the images and used a larger font size. For all figures we had to take into consideration that the maximum width of a two-column figure is 12cm and a one-column figure is 8.3 cm, which is predefined by the LaTeX template of SE. This limits the overall size of the figures. |
| 44 | | | Figure 1: text in figure too small | X | We used a larger font size. |
| 45 | | | Figure 2: in caption change 'yellow lines' to 'light green lines' | X | We changed 'yellow lines' to light green lines'. |
| 46 | | | Fugure 4: annotation for printed version too small, especially street names which are referred to in text | X | We replaced the entire image and used a larger font size. |
| 47 | | | Figure 7: upper part too small; the fold is very similar for each line and might not be as interesting as the elevation | X | We removed the fold and used a larger font size for the elevation. |
| 48 | | | Figure 8: this figure is certainly worth being shown as big as figure 9, maybe in the same manner | X | We changed the figure in accordance with the suggestion of the reviewer. |
| 49 | | | Figure 10: missing description of red circles | X | We added a description of the red circles in the caption. |
| 50 | | | Figure 10 and 11: A seismic interested person would certainly like to see the seismic data in a larger display. Both figures could gain from: - not showing all data down to 1200ms or 150m - omitting some of the repeated time and depth axis and annotations - less gaps between the panels. | X | Now the images of Figures 10 and 11 are bigger. We shortened the sections down to 1000 ms instead of imaging 1200 ms, and we omitted two of the repeating time-/depth-axes. |
| 51 | | Kammann: | Figure 11: I personally think that the interpreted results should be overlaid on the seismic. I find it very hard to transfer the faults and especially the depressions from 11c to 11b and therefore hard to follow the interpretation at all. | X | In the first version of the manuscript, Figure 11 was shown in the same manner you suggested, but after showing it to some neutral colleagues, all of them agreed that the interpretation is not clearly visible due to the seismic in the background. After some discussions we decided that the best style to show the interpreted results is as it is now. |
| 52 | | | Figure 1: could be more clear and maybe simplified, especially the legend. The reference LBEG (2015) is not in the reference list. The caption is unclear. Also the borehole information used for interpretation of the shear-wave data differ a lot from the average stratification. | X | The reference for 'LBEG 2015' is in the reference list, but it was written out. Now we added the abbrevation in the bibliography.
The legend is already simplified. It shows only the abbrevations of the geological units seen on the geological map.
The average stratification on the right covers the entire area of the geological map and was included in order to give a general overview of the stratification in the area. The borehole stratification gives detailed information about the geological units at the borehole location, which is located in profile S1. Variations are normal and depend on the complexity of the local geology. |
| 53 | | | Figure 7: compares 4 shots of the different profiles in means of noise and surface conditions. Which processing steps were applied to these shots? | X | The only processing steps applied to the data are vibroseis correlation, an AGC, a bandpass filter around the sweep frequency range and amplitude normalization. We added this information to the caption. |
| 54 | General | Anonymous: | While it is interesting and might for some readers be appealing, some parts of the manuscript are written in great detail and a bit repetitive. The manuscript could gain much from more conciseness. | X | We shortened some parts of the manuscript a little bit. |
| 55 | | | Please be careful with the use of 'seismics' as opposed to seismic. 'Seismics' it is not a noun and often only used in the informal language or colloquial speech (probably check with the desk editor in which way it is used in the journal). | X | We changed 'seismics' to 'seismic'. |
| 56 | | Kammann: | The numbering of the figures, parenthesis and use of capital letters should be checked in the text. | X | We checked the numbering, the parenthesis and use of capital letters. |
| 57 | | | There are some inconsistencies writing S/N ratio or signal-to-noise ratio in the text, please check. | X | We checked for consistent spelling. |
| 58 | | | Please also check that the terms "reflector" and "reflection" are used correctly. | X | We checked the text for the correct use of the words 'reflector' and 'reflections'. |
| 59 | | | The term Tertiary is not used anymore and should be substituted by Paleogene or Neogene (e.g. Page 3 Line 29 and following, Figure 1). | X | We replaced Tertiary with Paleogene and Neogene. |

[revised manuscript text omitted]
  the registration of converted waves. These waves were additionally suppressed by vertical stacking of records with opposite polarization during data processing. As a result this survey configuration enables straight forward data processing. The slower seismic velocities of SH waves enable images of higher resolution than using P waves  (e.g. Dasios et al., 1999; Inazaki, 2004). A further advantage is the autonomous suppression of surface Love waves, which occurs if the first subsurface layer is of higher

15  seismic velocity than the second layer, which is often the case on paved or compacted roads.

    The electro-dynamic micro-vibrators ELVIS 6 and ELVIS 7 (Fig. 6) are basically comparable and are used to generate horizontally-polarized shear waves (Polom, 2003; Druivenga et al., 2011; Krawczyk et al., 2012). ELVIS 6 was used on profile S1, and ELVIS 7 was used on profiles S2, S3 and S4 due to technical problems with ELVIS 6. The most important advantage of these small sources is the usability in urban areas  (Krawczyk et al., 2013), such as

20  the medieval center of Bad Frankenhausen, which is extremely built-up.

[revised manuscript text omitted]